# Overcoming evanescent field decay using 3D-tapered nanocavities for on-chip targeted molecular analysis

Shailabh Kumar [1,5], Haeri Park[1,2,5], Hyunjun Cho[3], Radwanul H. Siddique [1,2], Vinayak Narasimhan[1], Daejong Yang[1] & Hyuck Choo[1,2,3,4 ✉]

Enhancement of optical emission on plasmonic nanostructures is intrinsically limited by the distance between the emitter and nanostructure surface, owing to a tightly-confined and exponentially-decaying electromagnetic field. This fundamental limitation prevents efficient application of plasmonic fluorescence enhancement for diversely-sized molecular assemblies. We demonstrate a three-dimensionally-tapered gap plasmon nanocavity that overcomes this fundamental limitation through near-homogeneous yet powerful volumetric confinement of electromagnetic field inside an open-access nanotip. The 3D-tapered device provides fluorescence enhancement factors close to 2200 uniformly for various molecular assemblies ranging from few angstroms to 20 nanometers in size. Furthermore, our nanostructure allows detection of low concentration (10 pM) biomarkers as well as specific capture of single antibody molecules at the nanocavity tip for high resolution molecular binding analysis. Overcoming molecule position-derived large variations in plasmonic enhancement can propel widespread application of this technique for sensitive detection and analysis of complex molecular assemblies at or near single molecule resolution.

[1] Department of Medical Engineering, California Institute of Technology, 1200 E. California Blvd., MC 136-93, Pasadena, CA 91125, USA. [2] Image Sensor Lab, Samsung Semiconductor, Inc., 2 N. Lake Ave. Ste. 240, Pasadena, CA 91101, USA. [3] Department of Electrical Engineering, California Institute of Technology, 1200 E. California Blvd., MC 136-93, Pasadena, CA 91125, USA. [4] Imaging Device Lab, Device & System Research Center, Samsung Advanced Institute of Technology (SAIT), Suwon 16678, Republic of Korea. [5]These authors contributed equally: Shailabh Kumar, Haeri Park. ✉email: hyuck.choo@samsung.com

While enhancement of optical signals such as fluorescence using plasmonic nanostructures promised breakthroughs in areas such as single molecule fluorescence-driven DNA sequencing[1,2], rapid disease detection[3–5], as well as observation of biological reactions[6,7], their widespread application towards bioassays has remained lacking. One major reason for the lack of applicability of plasmonic fluorescence enhancement remains the wide variability in enhancement efficiency for molecular assays. Fluorescence enhancement obtained from plasmonic nanostructures is intrinsically dependent on both the efficiency of electromagnetic (EM) field confinement at the plasmonic hotspot as well as the distance between the optical emitter (fluorophore) and the plasmonic hotspot[8–12]. These fluorophores are typically attached to biorecognition elements such as antibodies or nucleic acid aptamers that recognize and specifically bind to other target molecules. Therefore the position or distance of light-emitting fluorophores with respect to a plasmonic hotspot is dependent on size as well as number of the molecules within the biomolecular complex, and can drastically alter the plasmonic enhancement of the emitted signal due to changes in the radiative and non-radiative field components[11–14]. Inconsistent or weak enhancement of signal due to these variations limits both the accuracy and efficiency of molecular binding analysis on chip. While several reports have discussed novel nanoscale geometries that improve the confinement of EM fields leading to strong fluorescence enhancements[15–18], engineering a hotspot that resolves the distance challenge between the emitter and the nanostructure surface has remained elusive.

Powerful fluorescence enhancement within a nanostructure independent of variation in molecule size and position can be expected to rely on several important factors: (a) strong confinement of electromagnetic field (b) powerful coupling of the emitter to the field for enhancement of emission and (c) a hotspot geometry, which generates uniform electromagnetic field distribution. At the same time, the hotspot geometry needs to be large enough for commonly used protein–protein binding assays (i.e. larger than antibodies, ~15 nm). Metal–insulator–metal (MIM) structures utilizing surface-plasmon-polariton (SPP) propagation have been known to enable efficient confinement of EM energy[19–22]. Specifically, waveguides with a 3D taper that rely on adiabatic compression of the SPP mode inside the MIM gap have been shown to provide extreme volumetric nanoscale confinement of EM fields[23–27]. While extremely promising for energy confinement and transfer, the potential of 3D-tapered designs for coupling with fluorescent emitters and bioassays had remained unrealized primarily due to closed monolithic structures, which prevented molecular integration with these devices. Furthermore, as electromagnetic field intensity on a plasmonic surface is maximum at the metal–dielectric interface, close packing of multiple metal–dielectric interfaces such as in very thin MIM gaps, can result in integration of these multiple field profiles within the gap creating a more homogeneous field distribution. Therefore, a 3D-tapered structure provides unutilized potential toward these goals, for enabling confinement of a large amount of incident electromagnetic energy into a tiny MIM gap.

In order to take advantage of the previously known as well as unexplored abilities of 3D-tapered MIM devices, we design and fabricate a fluidic channel-like 3D-tapered gap plasmon nanocavity, allowing ready access of the nanostructures to molecules in solution (Fig. 1a). We demonstrate that a 3D-tapered gap plasmon nanocavity can overcome a long-standing limitation for plasmon-enhanced fluorescence demonstrating powerful emission enhancement independent of the size or position of the molecules within the nanocavity. The 3D-taper results in confinement of the electromagnetic field collected throughout the body of the device into a tiny cavity (~3300× smaller in volume)

with a $|\mathbf{E}|^2$ enhancement close to 500. Furthermore, our analysis reveals that the 3D-taper geometry improves the coupling of molecular emitters to the electromagnetic field, delivering up to 28% improvement in the radiative decay rates and thus leading to even stronger enhancement of fluorescence. We specifically trap and observe single antibody molecules or arrays of molecular assemblies within the device, as well as detect low concentration (10 pM) protein molecules diffusing in solution. Significantly, optimizing the taper angle and tip geometry of the device results in 40× improvement in uniformity of the electromagnetic field volume compared with a bowtie nanoantenna. Combination of the strong electromagnetic confinement, powerful coupling of the emitter to the confined field and a homogenous electromagnetic field volume result in experimental enhancements of ~2200 compared with glass chips for molecular heights ranging from few angstroms to 20 nm, which can be further extended to 50 nm using the current design approach. Overcoming the molecule placement limitation for plasmonic enhancement of fluorescence such as presented in this manuscript can allow this technique to be reliably and widely applicable for a broad range of biological assays including complex molecular assemblies.

## Results

**Device fabrication and design optimization.** The 3D-tapered gap plasmon nanocavities were fabricated in gold- and silica-coated silicon substrates as shown in Fig. 1a, b. Detailed description of the fabrication process can be found in the methods section. The sidewalls that are ion-milled in the gold layer and the exposed top surface of the $SiO_2$ base, together form a fluidic MIM nanocavity that tapers vertically and laterally into a nanoscale tip (Fig. 1c). The hydrophilic $SiO_2$ base attracts fluid into the 3D-tapered nanocavity channel, promotes efficient molecular delivery, and provides surface site-specific molecular binding. Repeating the fabrication process, we also produced 3D-tapered gap plasmon nanocavity arrays with 20-nm wide and 500-nm long tips integrated onto a larger fluidic channel as shown in the scanning electron microscope images of Fig. 1e–g.

The design of the device was optimized for (a) efficient coupling of excitation light into the device body, (b) optimal transversal confinement of EM field- high and uniform $|\mathbf{E}|^2$ of the guided mode- through the taper, and (c) efficient longitudinal confinement of EM field at the tip. We chose 750-nm as the target wavelength for fluorophore excitation. Finite-difference time-domain (FDTD) simulations were utilized to accomplish efficient confinement of the fundamental anti-symmetric (AS) SPP mode at the tip of the 3D-tapered gap plasmon nanocavity. In the new design that allows hotspot access to fluids, a pair of Au walls separated horizontally on the $SiO_2$ substrate support the AS mode whose electric field is aligned parallel to the substrate (Supplementary Fig. 1a, b). Parameters were set to $w_{body} = 150$ nm, $h_{body} = 150$ nm, and $l_{body} = 3$ μm in order to accomplish efficient coupling of excitation light and low-loss guidance of the AS mode inside the body with high efficiency (Supplementary Note 1).

The taper angle ($\alpha$) was set to 20° in order to achieve efficient transversal confinement of EM field, which results in large and a uniform $|\mathbf{E}|^2$ profile at the tip (Supplementary Note 2, Supplementary Fig. 2). Figure 2a shows the profiles of EM energy density $u$ at the cross-sections of the body (top) and tip (bottom) at $\alpha = 20°$. The total transversal EM energy stored inside the body ($U_{A\_body}$) is efficiently confined inside the tip with minimal loss ($U_{A\_body} \sim U_{A\_tip}$), which significantly increases the average transversal EM energy density $\overline{u}_A$ at the tip (Fig. 2b, Supplementary Note 2). The 3D-tapered gap plasmon nanocavity showed greater confinement of EM energy (greater $\overline{u}_A$ at the tip) than the

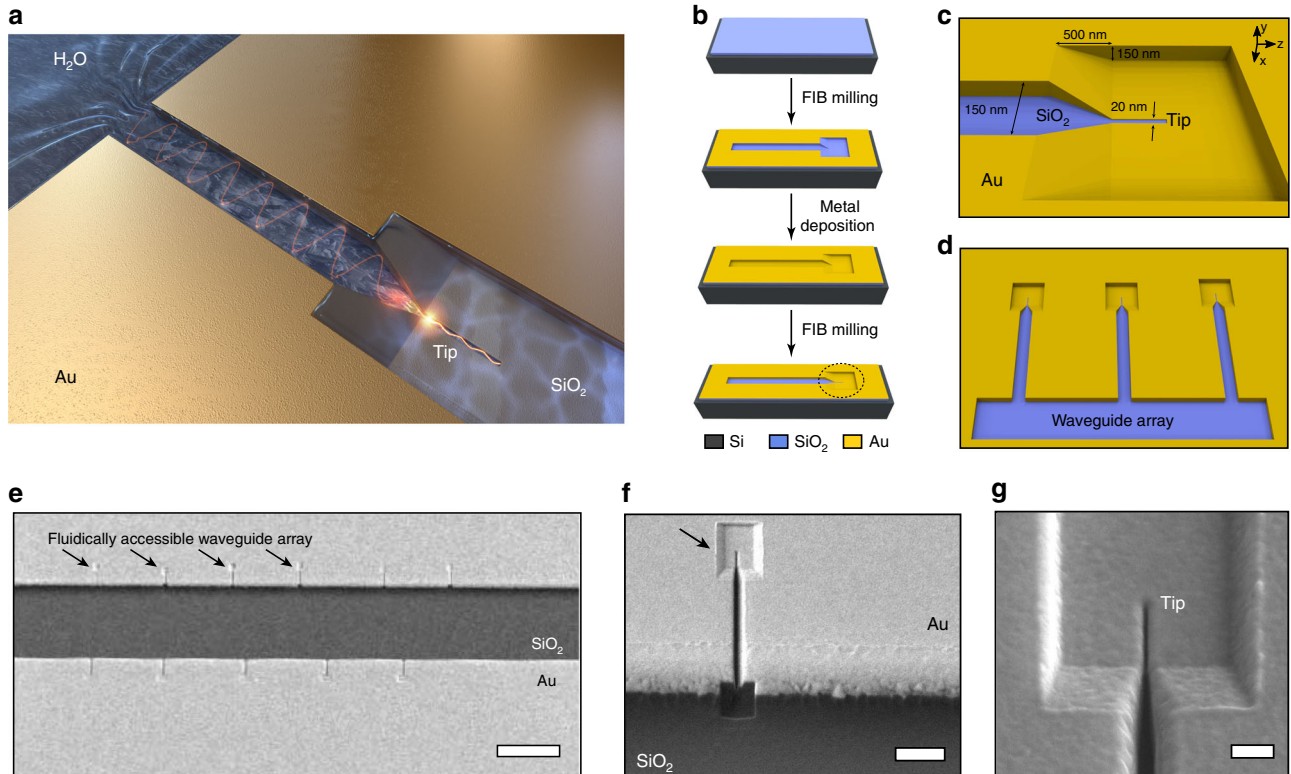

**Fig. 1 Fabrication of 3D-tapered gap plasmon nanocavities. a** Overview image of the device with plasmonic wave propagation and confinement at the end of three-dimensionally tapered tip. Open cavity allows fluidic delivery and surface functionalization enabling molecular capture. **b** Device fabrication using a silicon wafer with 1 μm thick thermally grown silicon dioxide, gold deposition, and FIB milling. **c** Zoomed-in view of the 3D-tapered nanocavity tip. **d** Schematic showing an array of 3D-tapered nanocavities with a hydrophilic silica base. **e** SEM image showing fabricated arrays of 3D-tapered nanocavities with hydrophilic silica base. Scale bar is 25 μm. **f** A single 3D-tapered nanocavity. Scale bar is 2 μm. **g** Sharp 3D-tapered naocavity tip. Scale bar is 200 nm.

2D-tapered nanocavity and the tip-only structure (MIM structure without a taper) of the same tip size due to a larger cross-sectional area of the body being capable of storing greater $U_{A\_body}$ (see "Methods"). This extreme confinement of EM energy through the 3D taper suppresses evanescent-field type decay inside the $20 \times 50$ nm$^2$ tip (MIM gap), which is generally observed in high-$|\mathbf{E}|^2$ plasmonic structures such as a bowtie nanoantenna[15,28] or tip-only structures (MIM waveguides without a taper)[29] (Fig. 2c, d and Methods). The optimized 3D taper achieved four times greater $\overline{u}_A$ and 40× improvement in uniformity $\sigma_{|\mathbf{E}|2}$ than a bowtie nanoantenna of the same gap size (Fig. 2e). All of the gap plasmon structures studied in the FDTD simulations -3D-, 2D-tapered nanocavities, and the tip-only structure showed a trend that a greater $\overline{u}_A$ enabled a more uniform hotspot (a smaller $\sigma_{|\mathbf{E}|2}$) and greater average $|\mathbf{E}|^2$ ($\overline{|\mathbf{E}|^2}$). As a result, the optimized 3D-tapered nanocavity provided a large ($\overline{|\mathbf{E}|^2} = 230$), uniform ($\sigma_{|\mathbf{E}|2} = 0.05$) $|\mathbf{E}|^2$ profile at the tip along with an 11% net coupling efficiency. Experimental analysis of the taper angle and body width also matched the trend predicted by simulations, as maximal emission output was obtained from devices with taper angle $\alpha\sim 20°$ and $w_{body} = 150$ nm (Supplementary Fig. 3).

The volume and $|\mathbf{E}|^2$ profile of the hotspot can be further optimized for capture of targeted number of molecules by varying $l_{tip}$. A shorter $l_{tip}$ allows the coupled EM energy to be more densely packed longitudinally inside the tip, which results in greater $|\mathbf{E}|^2$ within a smaller hotspot volume. A 20-nm long tip provided a $\overline{|\mathbf{E}|^2}$ magnitude of 550 within a $20 \times 50 \times 5$ nm$^3$ hotspot (Fig. 2f, g, Supplementary Note 3, Supplementary Fig. 4). Based on the simulations, experimental results and

considerations, devices were fabricated with the aforementioned dimensions (as shown in Fig. 1) and fluorescence enhancement was studied.

**Volumetric optical confinement and biosensing**. We experimentally tested coupling of incident light into the nanocavity and gap plasmon-mediated volumetric confinement at the tips using surface-linked molecular layers and fluorescent labels. A molecular monolayer of biotin was assembled along the exposed silica surface of the 3D-tapered nanocavity, using silane-polyethylene glycol-biotin (SPB) as the reagent to form silane–silica covalent linkages (Supplementary Fig. 5a). Streptavidin linked with Alexa Fluor 750 (S-AF 750) was then used as a fluorescent label for detection of biotin in the tips, taking advantage of the well-known strong and highly-specific molecular interaction between biotin and streptavidin[30,31]. This two-step binding reaction allows the formation of a monolayer of fluorescently-tagged streptavidin on the silica surface along the length of the 3D-tapered nanocavity. We observed the capture of diffusing streptavidin molecules at high concentrations (1 μM) and resultant enhancement in fluorescence once a molecule binds to the tip region (Supplementary Fig. 5b, c). Detection of molecules diffusing in solution at micromolar concentrations has been an important target for biological tracking and has been demonstrated using nanostructures with zeptoliter detection volumes[16]. However, these detection methods lacked molecular specificity commonly required for bioassays. The silica base of our device serves as a targeted functionalization region inside the gap plasmon nanocavity, allowing specific capture of molecules diffusing in solution

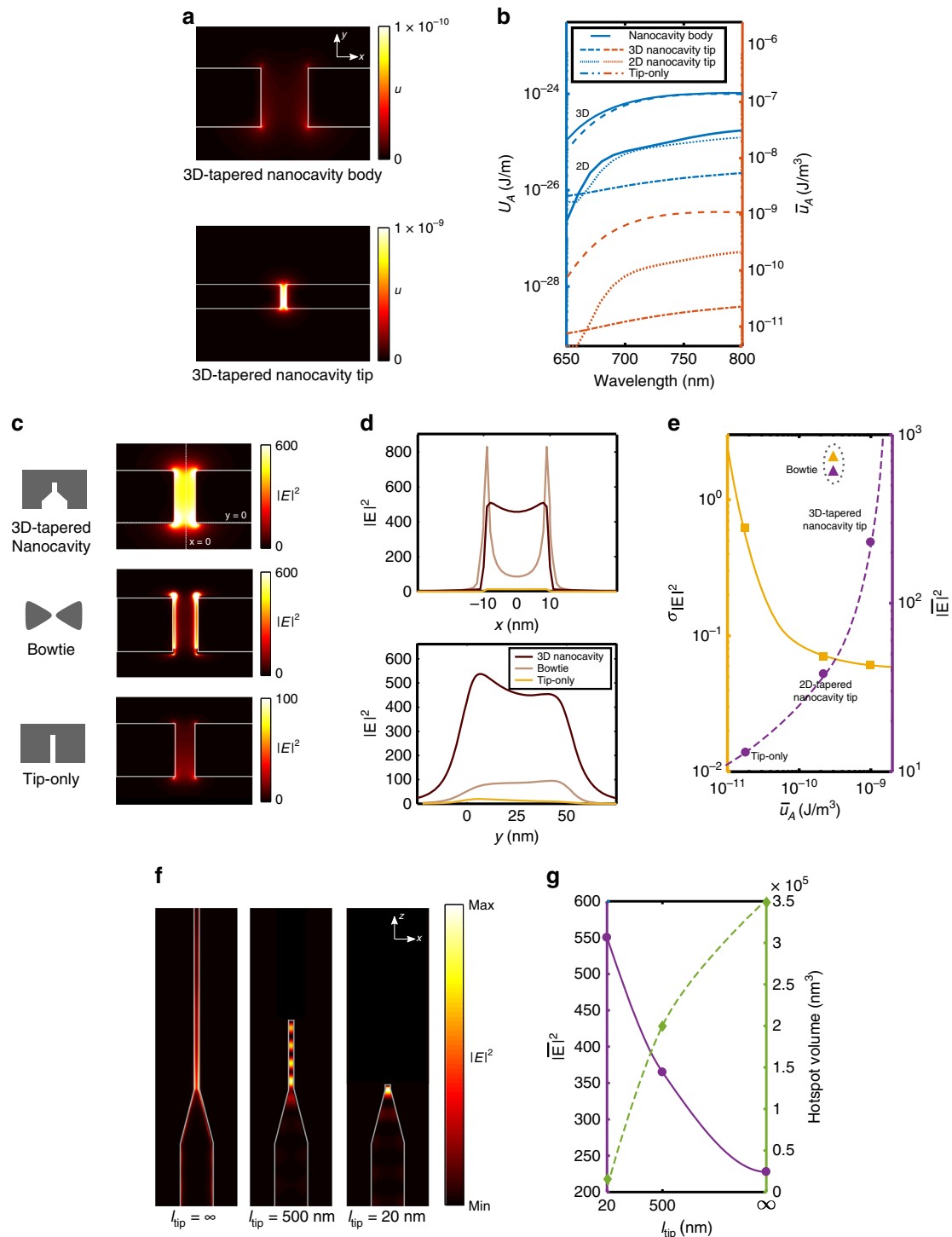

**Fig. 2 Design optimization of the 3D-tapered nanocavity using FDTD simulation. a** Comparison between the cross-sectional views of the EM energy density profiles in the body (top) just before the 3D-taper ($\alpha = 20°$) and at the tip (bottom). **b** The total transversal EM energy $U_A$ and average transversal EM energy density $\bar{u}_A$ of the body and tip of the 3D- and 2D-tapered nanocavities and a tip-only structure. **c** Cross-sectional view of the $|\mathbf{E}|^2$ profiles of the 3D-tapered nanocavity (top), a bow tie (middle), and a tip-only structure (bottom) of the same gap size (20 nm × 50 nm). **d** $|\mathbf{E}|^2$ inside the gaps of the structures at $y = 0$ (top) and $x = 0$ (bottom). **e** Hotspot uniformity $\sigma_{|E|2}$ and average $|\mathbf{E}|^2$ at cross-sectional areas of the 3D- and 2D-tapered nanocavities, a tip-only structure (circle), and a bowtie (triangle). **f** Top view of the $|\mathbf{E}|^2$ profiles in the devices with $l_{tip} = 20$ nm, 500 nm, and $\infty$. **g** $|\mathbf{E}|^2$ and hotspot volume at the tips of the devices with varied $l_{tip}$.

and differentiating them from the background through enhanced fluorescence signal.

After molecular binding, chips were washed to remove unbound molecules and interrogated using tail-end and full illumination modes (Fig. 3a, b). The tail-end illumination mode involves illumination of the back-end of the 3D-tapered nanocavity, whereas full illumination mode allows the complete device to be placed under illumination (Supplementary Note 4). Excitation light from a near-infrared light-emitting diode light source (750 nm) was incident on the 3D-tapered nanocavities and

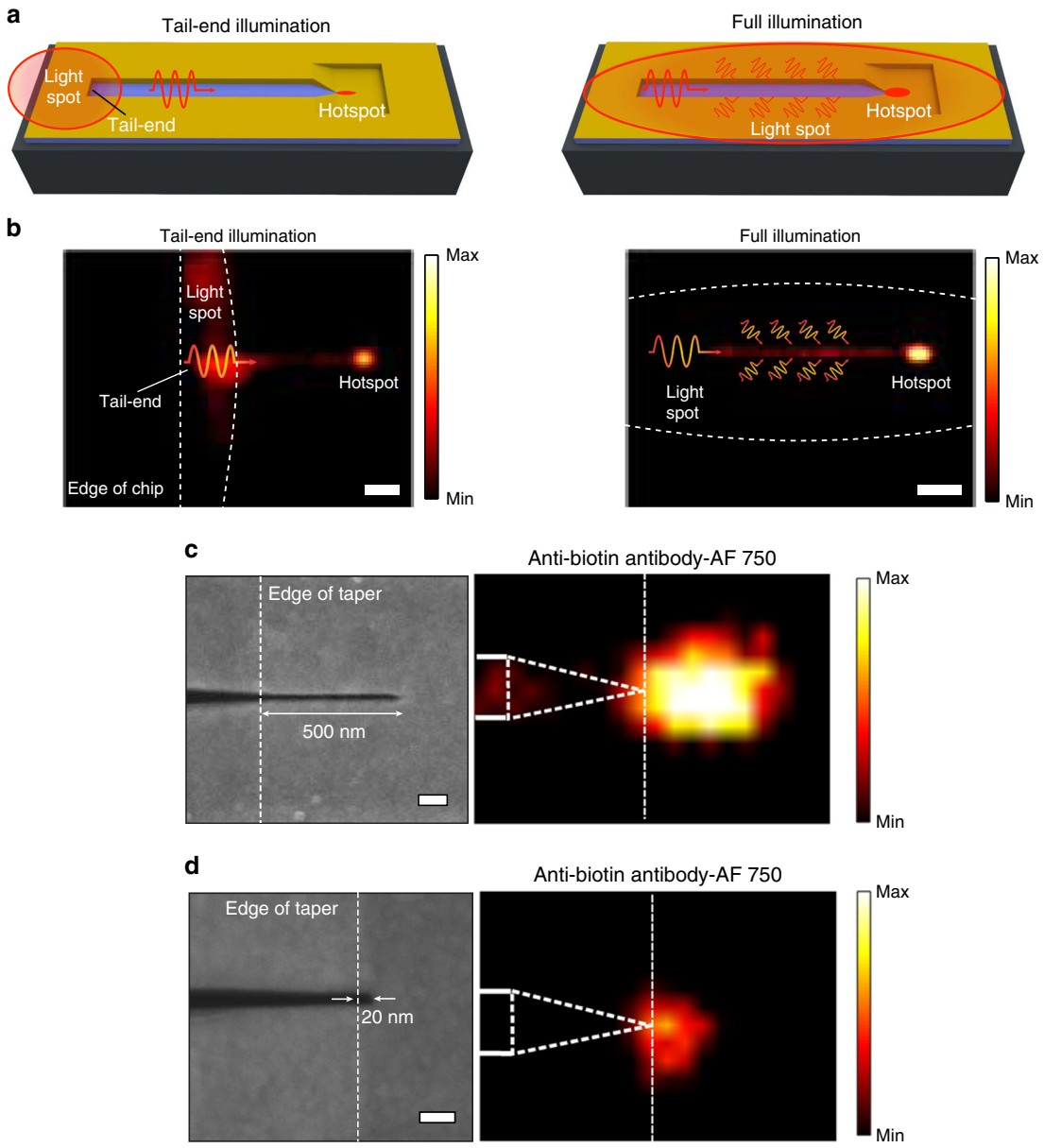

**Fig. 3 Molecular fluorescence enhancement and single molecule capture. a** Illustration and **b** fluorescence images of streptavidin capture inside the nanocavities using tail-end (left) as well as full (right) illumination modes. Hotspot is visible at the end of taper within the 20 nm-wide tip. Scale bars are 1 μm. **c** SEM image (left) of a 3D-tapered nanocavity with tip length 500 nm and fluorescence image (right) obtained from the tip after antibody immobilization. 1D-array of antibodies is expected within the tip. Scale bar is 100 nm. **d** SEM image (left) of a 3D-tapered nanocavity tip with length 20 nm and fluorescence image (right) obtained from the short tip after antibody immobilization. Volumetric-limitation enables specific capture of single antibody at the end of the tip. Scale bar is 100 nm.

a visible hotspot in the form of enhanced fluorescence was observed from the sub 20-nm tip region as a result of volumetric field confinement at the tips (Fig. 3b). Due to improved coupling and light collection through the body of the device, about an order of magnitude improved intensity (~9×) at the tip was obtained using full illumination mode as compared with tail-end illumination mode, which agrees with the calculation based on the FDTD simulations (~10×) (Supplementary Fig. 6). The role of the 3D-tapered nanocavity body towards fluorescence enhancement was further verified by fabricating dimensionally-varying structures (Supplementary Note 5). Fluorescence intensity comparison between the samples indicated that stand-alone taper and tip structures have a significantly weaker performance for optical molecular analysis as compared with the complete device

(Supplementary Fig. 7). This observation can again be attributed to improved coupling efficiency for the full device (Supplementary Fig. 8), in addition to light collection through the device body. For subsequent molecular detection and analysis of plasmonic enhancement, full illumination mode was implemented.

Limit of detection for biomolecules on the device was examined using two types of sensing experiments. First, detection of low concentration protein molecules in solution (10 pM–1 nM) was performed, testing device suitability for on-chip diagnostics of rare disease-specific biomarkers (Supplementary Fig. 9). The detection limit in this case is governed by diffusive transport of molecules to the plasmonic hotspot and can be accelerated by improving molecule transport using

convective flow, magnetic or dielectrophoretic trapping. Second, we examined the capture and detection of individual or small array of molecules at the hotspot, while high concentration of molecules were present in solution. User-controlled analysis of single or small array of molecules remains an important target for high-resolution analysis of protein function and behavior[32–34]. This is especially important in cases where ligands and biomolecules are physiologically present at higher (μM–mM) concentrations[35]. We control the number of molecules captured within the tips by altering the tip length. After formation of the biotin monolayer within the 3D-tapered nanocavities, we utilized an anti-biotin IgG primary antibody (tagged with DyLight 755 ~spectral response similar to AF-750) for performing fluorescence assays[36]. In order to compare effect of tip length on capture of molecules, we performed these antibody binding experiments using devices with $l_{tip} = 500$ nm as well as $l_{tip} = 20$ nm (Fig. 3c, d). The dimensions of a 3D-tapered nanocavity with a short tip ($l_{tip} = 20$ nm, $w_{tip} = 20$ nm) compared with those of an IgG antibody (length ~15–20 nm)[36,37] indicate that a single antibody should be specifically trapped within the tip region. We performed atomic force microscopy (AFM) which demonstrated uniform monolayer of PEG-biotin and antibodies on flat silica surfaces and indicated that only one antibody should specifically occupy the 20 nm × 20 nm tip (Supplementary Fig. 10). We obtained 1.5% variation between expected and measured fluorescence intensity for a single molecule at the short tip, further indicating presence of a single antibody at the short tip-end. The calculation was based on integrated fluorescence intensities from the tips and simulation-predicted $|\mathbf{E}|^2$ profiles (Fig. 2a) along with an assumption of packed molecular arrangement within the tips.

**EM field decay-resistant plasmon-enhanced fluorescence**. We analyzed the enhancement of fluorescence experienced by an emitter within the tip, as compared with a non-enhancing substrate (silica or glass surface). As discussed previously, the tip region of the device exhibits a hotspot with uniformly distributed, highly confined electromagnetic field (Fig. 4a). The emission enhancement was calculated using FDTD simulations (Fig. 4b, c), where the net fluorescence enhancement experienced by a fluorophore is defined as a product of EM field intensity and quantum yield gain (Supplementary Note 6, Supplementary Figs. 11, 12). As discussed in detail in Fig. 2 and Supplementary Fig. 2e, the 3D-tapered gap plasmon nanocavity provides superior $|\mathbf{E}|^2$ uniformity within the gap compared with conventional MIM structures. Our analysis also shows that the 3D-tapered gap plasmon nanocavity allows greater radiative decay rate that results in up to 28.2% enhanced quantum yield gain along the width of the channel as compared with the tip-only structure, thereby enabling ~500× greater net fluorescence enhancement (Supplementary Fig. 13, Fig. 4b, c). For a 500 nm long tip, the expected fluorescence enhancement is >1000 for about 70 % of the channel width. The full-width half-maximum (FWHM) covers 95.5% of the $x$-axis with indication of fluorescence quenching very close to the metallic sidewalls (Fig. 4b). Fluorescence enhancement is very uniform (~1000) along the $y$-axis and does not decline below half of the maximum enhancement throughout the channel height (Fig. 4c). This is especially important as variation along the y-axis represents the increase in fluorophore height from the bottom silica surfaces. This can be a result of the size of molecule the fluorophore is attached to or based on number of molecules in the assembly. The simulations indicate that this variation should have minimal impact on the enhancement experienced by the emitter, which has been a long-standing challenge for plasmonics-enhanced fluorescence.

In order to experimentally verify the optimized fluorescence enhancement response predicted in Fig. 4b, c, we performed tests using diversely-sized molecules including dye molecules, aptamers, smaller proteins and antibodies. These molecules were specifically bound to the 3D-tapered nanocavity, and fluorescence enhancement was analyzed compared with non-structured SiO$_2$ control samples. While the dye molecules were covalently bound to the bottom silica surface, all other molecules were specifically bound to their respective recognition agents or antigens preassembled on the surface (Methods and Supplementary Note 7). The expected height of the fluorophore from the silica base of the nanocavity is indicated in Fig. 4d and varies from <1 nm to ~20 nm. The experimentally calculated enhancement performance of the device showed a uniform response for various molecular shapes and heights, which agrees with the simulation results shown in Fig. 4c. We expect this device to maintain the same enhancement for molecular assemblies with height up to 50 nm, which is defined by the height of the tip for this device. The level of enhancement was dependent on the tip length as expected. For tips with length 500 nm, the enhancement was close to 950 whereas shorter tips ($l_{tip} = 20$ nm) provided higher enhancement (EF~2200) due to stronger $|\mathbf{E}|^2$ enhancement at the tip region for diverse molecular sizes (Fig. 4e, Supplementary Fig. 14). These results matched the trend and values predicted earlier by simulation-based analysis (Supplementary Fig. 3c, Supplementary Note 8).

As fluorescence enhancement is dependent on the quantum yield of the fluorophore used, we also used another metric, enhancement figure of merit—which normalizes the enhancement with respect to the dye quantum yield and allows comparison of device performance to other nanostructures (Supplementary Table 1)[18]. The device showed an enhancement figure of merit close to 260, which is one of the highest values obtained for fluorescence enhancement obtained using plasmonic nanostructures, and uniquely provides this enhancement independent of molecule size within the nanocavity tip.

## Discussion

We have demonstrated 3D-tapered gap plasmon nanocavities, which provide one of the highest enhancement of fluorescence obtained by plasmonic nanostructures (EF: ~2200 with figure of merit ~260), independent of the size of the molecular assemblies used in the assay. Overcoming molecule size and placement-dependent extreme variation in plasmonic fluorescence enhancement has been a major challenge restricting widespread application of this method in bioassays. The nanostructure geometry presented in this work demonstrates a way to overcome this limitation thereby improving the consistency and range of plasmon-enhanced emission for diversely sized assembly of molecules. Simultaneously, we also demonstrate capture and visualization of single antibodies at the tip as well as sensing of proteins at low concentrations (10 pM). These advantages can be readily transferred towards applications such as highly sensitive biosensing using molecular labels of varying sizes and analysis of single molecules or tightly controlled arrays of molecules for protein orientation, protein function, and biological polymer formation studies[32–34]. While the device geometry promises several benefits, weaknesses of the current fabrication process include low throughput using focused ion beam lithography and high footprint of the device compared with smaller nanostructures. Future improvements can target the replacement of current fabrication process with wafer-scale methods including nanoimprinting, e-beam lithography, and template stripping for reproducible manufacturing of nanocavities in combination with anisotropic etching methods for the tapered portions. We may

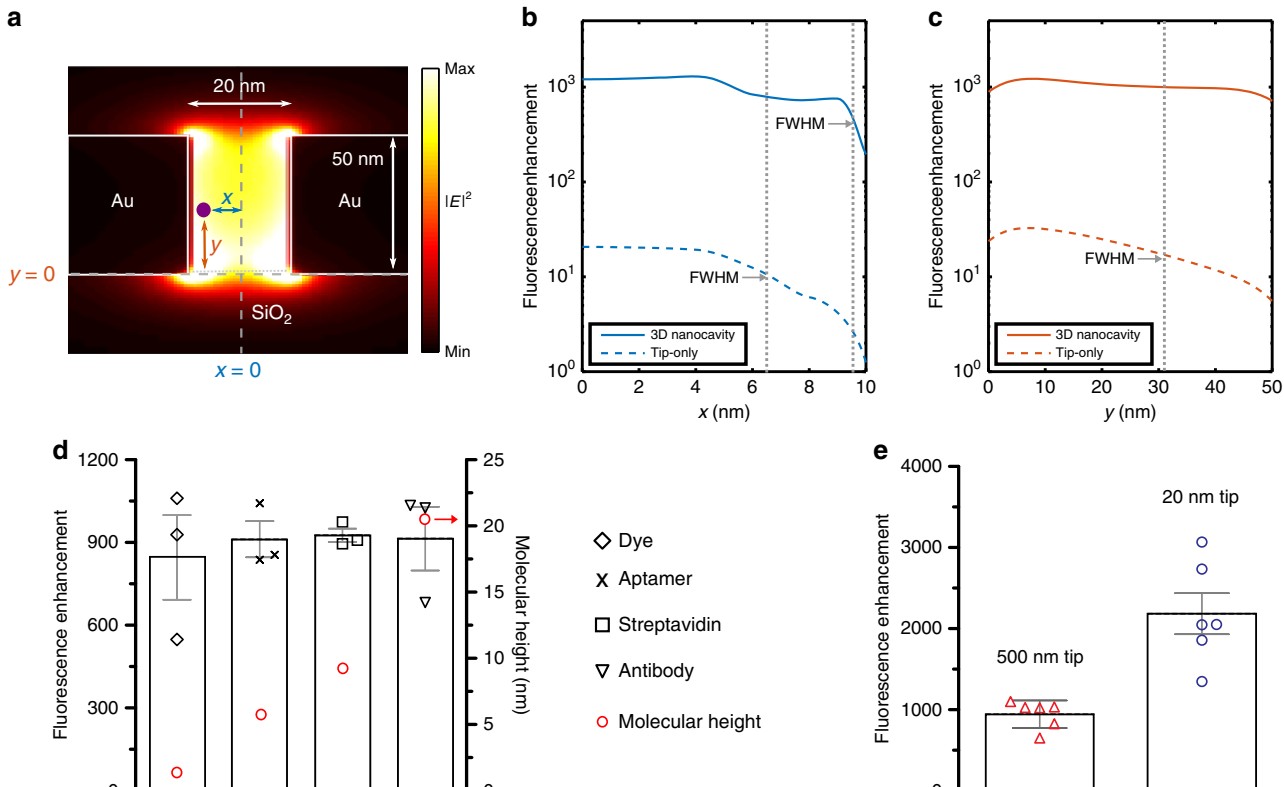

**Fig. 4 Overcoming electromagnetic field decay for molecular analysis. a** Cross-section of the tip at the end of taper showing EM field intensity and placement of a fluorophore with distance along the $x$ and $y$ axis for a 500 nm long tip. Variations in net fluorescence enhancement, calculated as a product between the EM enhancement and quantum yield gain, of the 3D nanocavity (solid line) and tip-only structure (dashed line) as a function of molecular position along the **b** $x$- and **c** $y$-axes. Metal-induced quenching effects are visible near the metallic sidewalls of both structures. The 3D nanocavity shows uniform fluorescence enhancement for about 95.5% (FWHM) of the $x$-axis and 100% of the $y$-axis. **d** Experimentally obtained enhancement factors calculated for various molecular assemblies using a tip with length 500 nm, demonstrating highly-enhanced fluorescence from all samples as a result of their placement within the plasmonic hotspot. The estimated height of the molecules from the silica surface has been shown. The near uniform mean enhancement indicates suitability towards using molecular assemblies of different heights, overcoming evanescent field decay typically associated with plasmonic nanostructures. Plot shows datapoints, mean and s.d. for three devices at each condition. **e** Increase in mean enhancement factor with decrease in tip length to 20 nm. Plot shows datapoints, mean and s.d. for six devices at each condition.

also see further improvement in device performance after replacing ion beam milling with alternative methods mentioned above, which are known to yield smoother device surfaces, and have shown such improvements in the past[18]. The presented device design can also provide advantages in other areas of nanophotonics for optical confinement, data-transfer, quantum optical communication, and molecular sensing in mid-infrared and terahertz domains.

## Methods

**3D-tapered nanocavity fabrication.** Single-side polished silicon wafers with thermally grown $SiO_2$ (thickness: 1 μm) were purchased from University Wafers, Boston, USA. E-beam evaporation was used to deposit 50 nm gold (Au) on the wafers. 3D-tapered nanocavity patterns were milled through the gold and silica using a FEI Nova 600 dual beam system as shown in Fig. 1. Au (50 nm) was deposited again using e-beam deposition. Second round of milling was performed using Nova 600 to remove gold from the bottom of the substrates, exposing the silica and to mill the tip.

**Simulations.** We used a commercial software developed by Lumerical Inc. for the FDTD analyses. In all analyses, the mesh size was 1 nm and uniform throughout the device area. For the $|\mathbf{E}|^2$ and uniformity analyses, a 750-nm dipole source was placed at the tail-end of the 3D-tapered nanocavity body and the intensity of the guided AS mode was monitored across the cross-section of the tip. $|\mathbf{E}|^2$ at each $\alpha$ was calculated by averaging the $|\mathbf{E}|^2$ profile over the cross-sectional area ($20 \times 50$ $nm^2$) of the tip. $\sigma_{|\mathbf{E}|^2}$ was calculated by normalizing the $|\mathbf{E}|^2$ enhancement profiles (setting 1 as the highest) and calculating 2D standard deviation of the profiles over

the cross-sectional area. 2D-tapered nanocavity was designed to provide only lateral confinement (body cross-sectional area: $150 \times 50$ $nm^2$) along with the same taper length of 500 nm and tip cross-sectional area ($20 \times 50$ $nm^2$) as the 3D-tapered nanocavity. The tip-only structure was a simple MIM waveguide without a taper that provided the same cross-sectional area ($20 \times 50$ $nm^2$). The bowtie antenna was a set of two equilateral Au triangles (side length 140 nm, thickness 50 nm) separated by 20 nm on a 3-nm thick Cr layer on top of a 25-nm thick ITO[15,28], which provided the same cross-sectional area at the gap ($20 \times 50$ $nm^2$). For the coupling efficiency into body analyses (Supplementary Figs. 6, 8), a 1.4-NA Gaussian source centered at 750-nm was focused onto either the tail-end or varied longitudinal locations ($z$) throughout the body. In both cases, transmitted power through the cross-section was calculated at each location of incidence. For the fluorescence enhancement analysis, a 750-nm dipole source was placed inside the tip and transmitted power through a closed box enclosing the dipole and tip was monitored in order to obtain radiative and non-radiative decay rates (see Supplementary Note 6 and Supplementary Figs. 11,12).

**Dye (AF-750) binding.** 3-Aminopropyl-triethoxysilane (APTES) was obtained from Sigma-Aldrich, USA. The nanocavity chips were cleaned with acetone, methanol, and isopropanol prior to binding experiments. A 1% APTES solution was prepared in toluene (anhydrous) and allowed to bind on the chips for 30 min. This step functionalizes the silica surface with amino groups available for further binding. The substrates were washed with toluene to remove weakly bound APTES molecules. The chips were baked at 110 °C for 30 min and then cleaned again with DI water for 15 min. The samples were then dried with nitrogen. Alexa Fluor 750 NHS ester was purchased from Thermofisher Scientific USA and dissolved in DMSO (1 mg/ml). The solution was added to the substrates with stirring and dye molecules were allowed to bind to the functionalized substrates for one hour. The chips were then cleaned well with DMSO and water. A drop of water was placed on the chips, covered with a coverslip, and then imaged.

**Aptamer sensing**. Insulin from bovine pancreas was purchased from Sigma Aldrich (USA). Insulin was mixed in PBS at a concentration of 10 μM. Solution was added to the chips and left undisturbed for an hour to allow physisorption of the peptide hormone to the surfaces. BSA (0.1%) was then added to the chips and allowed to interact with the surface to account for any nonspecific binding. An insulin-binding aptamer (sequence: 5′ GGT GGT GGG GGG GGT TGG TAG GGT GTC TTC 3′) with AF-750 conjugated to the 5′ end was obtained from IDT technologies (St. Louis, USA). Aptamer was dissolved at a concentration of 1 μM in a folding buffer (Tris: 10 mM; MgCl$_2$·6H$_2$O: 1 mM; NaCl: 100 mM; pH: 7.4) and added to the substrate for an hour. The samples were washed with folding buffer after an hour and then imaged.

**Surface biotinylation**. SPB (MW: 1000 and 5000) were purchased from Laysan Bio Inc. (USA). For surface functionalization, SPB powder was dissolved in 50% ethanol: DI water. This solution was added to substrates with exposed silica surface and left undisturbed for an hour. The substrates were then washed with DI water.

**Streptavidin binding**. Streptavidin Alexafluor 750 conjugate (SAF-750) was purchased from Thermofisher Scientific (USA). Streptavidin in PBS (pH 7.4, concentration: 0.1 mg/ml) was added to chips with biotin monolayer and left undisturbed for an hour. The chips were then washed with DI water and imaged. A layer of water was maintained at all times on the chip through molecular functionalization and imaging. For Streptavidin bioassays, the concentration of streptavidin in PBS was varied while maintaining the duration of incubation (1 h).

**Antibody binding**. Anti-biotin antibody (Mouse monoclonal (Hyb-8), IgG1, MW: 244 kDa) conjugated with DyLight 755 fluorophore (response similar to Alexafluor 750) were obtained from Novus Biologicals (USA). The antibody was diluted in PBS at a concentration of 0.05 mg/ml and added to the chips. The solution was left undisturbed for 2 h. The samples were then washed with DI water.

**Atomic force microscopy (AFM)**. Topology and phase images of dried antibody monolayer on silica substrates were obtained on an AFM system (Bruker Dimension Icon, Santa Barbara, CA, USA) using a 100 μm long monolithic silicon cantilever (All-In-One-Al, NanoAndMore USA Corp (BudgetSensors), Watsonville, CA, USA). All the experiments were conducted under ambient laboratory conditions using tapping mode with a resonance frequency of about 350 kHz. Images were analyzed afterward by commercial software Nanoscope Analysis.

**Imaging**. Fluorescence imaging was performed using a Leica DMI 6000 widefield fluorescence microscope. Scanning electron micrographs were taken using FEI Nova 600 and 200 dual beam systems.

**Image analysis**. Images were analyzed using Fiji (ImageJ) software[38]. Graphs were created in Matlab and Graphpad Prism 5.01.

## Data availability

The data that support the findings of this study are available from the corresponding author upon reasonable request.

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

## Acknowledgements

Device fabrication was performed at the Kavli Nanofabrication Center at California Institute of Technology. Fluorescence imaging was performed at the Beckman Imaging center at California Institute of Technology. Funding for this research was provided by HMRI Investigator award and Samsung GlobalResearch Outreach (GRO) program.

## Author contributions

S.K., H.P., and H.C. conceived the study. S.K. performed the device fabrication and the molecular fluorescence experiments. H.P. performed the theoretical device design optimization and numerical simulations. HJ.C. and D.Y. helped establish device fabrication protocol. R.H.S. helped with numerical simulations and device optimization. V.N. assisted with data visualization and validation. S.K., H.P., and H.C. wrote the paper.

## Competing interests

The authors declare no competing interests.
