## [Peer Review File · Nature Communications]

Reviewers' comments first round:

Reviewer #1 (Remarks to the Author):

The manuscript by Kumar et al entitled “Overcoming Evanescent Field Decay Using 3D-Tapered Nanocavities for On-Chip Targeted Molecular Analysis” discusses how using an “open,” tapered, metal-insulator-metal waveguide enables the formation of a rather uniform, strongly enhanced electric field. Due to the “openness” of the cavity the enhanced electric field is accessible to e.g. analyte molecules and can therefore be used as a sensing platform. According to the Authors, the key breakthrough reported in the submitted manuscript is the creation of an enhanced electric field which is very uniform. Hence, biosensing with this device is not limited by the inhomogeneous field and the device works, according to the Authors, just as well for a short fluorescent molecule (tag) or a long one. The Authors first present a numerical analysis of the field enhancement in the device and then use fluorescent labels of different length/height to demonstrate uniform fluorescence enhancement within the whole MIM waveguide.

In general, I find little that is wrong with this manuscript/work. It is pretty well written and easy to follow/understand, however, certain statements are a little confusing and/or lack precision. In my opinion the only drawback of this manuscript is that the concepts which are its basis are already very well studied. There are two key parts to making this device – the energy concentrator (taper) and an appropriately sized final MIM waveguide – are not unique. Perhaps the combination of the above mentioned elements in this particular realization is unique, but there are no new physical concepts involved in this work.

The “overcoming of the evanescent field decay” is done using a well-studied MIM waveguide. Its “[non]evanescent field” in the final, narrow part is the same practically regardless of what how the energy is coupled into it, neglecting any transient effects. It is simply given by the geometry. Such MIM waveguides have been presented as biosensors before by, e.g. Dell’Olio et al (Design of a New Ultracompact Resonant Plasmonic Multi-Analyte Label-Free Biosensing Platform, *Sensors* 17, 1810 (2017)), as well as others.

I am puzzled by the “tip-length effect” phrase. The “cavity” (and here, in contrast to the Authors, I use “cavity” to refer only to the narrow, final part) is merely a MIM waveguide with a finite length so that a Fabry-Perot resonance can be used to modulate the field intensity in the hot spot (note, that the hot spot becomes confined in the third dimension).

Also, while I understand the reason behind using the 3D taper, I do not think that the tapered part is all that important in achieving “3 to 4 times greater u_A (field uniformity)” (as stated on page 4). The uniformity is ensured by the MIM waveguide, while the role of the 3D taper is to compress the incident field (from the large MIM waveguide).

Reviewer #2 (Remarks to the Author):

Reviewer Blind Comments to Author

Manuscript Number:

Title: Overcoming Evanescent Field Decay Using 3D-Tapered Nanocavities for On-Chip Targeted Molecular Analysis

Recommendation: Reject (Publish in a different journal)

This paper presents a 3D tapered nanocavity which provides the enhancement factor of 1100 and the 40-fold fluorescence enhancement uniformity compared to a conventional bowtie nanoantenna. They fabricated the 3D tapered nanocavity device by using focused ion beam (FIB) milling and gold deposition on the SiO₂/Si substrate. They also measured fluorescence signal (AF-750) from the biotin-streptavidin binding using the 3D tapered nanocavity under the near-infrared LED excitation. While the paper contains some of the necessary data, it is missing experimental results and discussion for why a 3D tapered nanocavity was made and how it is substantially better than all previous works.

1. The novelty of a 3D tapered nanocavity is not clear. The 3D tapered nanocavity structure is basically same as the previously reported 3D tapered metal-insulator-metal waveguide (Choo et al. Nature Photonics, 2012). Please clearly deliver the novelty of this structure (3D tapered nanocavity) compared to previous SPP or LSPR sensors in terms of device and performance (E-field enhancement, fluorescence signal enhancement, etc.).
2. There are parameter studies using the FDTD calculation in the supporting information. However, overall experimental studies are insufficient (Fig. 2 and Fig. 3). They only show simple experimental results for the 3D tapered nanocavity, the tip only structure, and the conventional bowtie nanoantenna. It would be better to show and discuss experimental results for the 3D tapered nanocavities with different physical dimensions as shown in the numerical results.
3. The device coated with biotin can capture the streptavidin high concentration. What is the limit of low concentration?
4. They demonstrated specific and single molecular capture using the nanocavity which has similar dimensions with the target molecule. In this paper, they capture the IgG antibody (length 15~20 nm) using the nanocavity with 20 nm x 20 nm size. If this single molecule capture depends on the size of

nanocavity and the target molecule, additional experimental results for different sizes (at least two or three) can support the principle of single molecule capture.

5. In addition, the authors should carefully proofread the manuscript. For example,

- Please insert scale bars for all figures of 2D E-field intensity profiles.

- In Figure 3, the alphabets (numbers) for sub-figures and descriptions don't match. In addition, it would be better to effectively display their results to place the results under the corresponding schematic illustrations for effective delivery of results.

- On page 4, Finite-Domain Time-Difference is incorrect terminology for FDTD. Finite Difference Time Domain (FDTD) is correct terminology.

- In Figure S4b, the direction of arrow is strange. It should be in the opposite direction.

The authors present an interesting concept to improve the fluorescent enhancement intensity using a plasmonic nanotapered cavity. The concept is novel, and will be of interest to the biosensing research community, and also to other branches of photonics research. The work is very thorough, and the results are convincing. However, the manuscript is not written very well, and a more balanced view of the importance of the work should be presented.

Comments regarding writing and presentation of results

Some of the examples about the writing and organization are described below

a) In page 3 of the manuscript the authors mentioned

“This engineered hotspot within the device tip was used for highly efficient and uniform fluorescent enhancement – the enhancement factor up to 1100 times and 40x improvement in uniformity compared to a bowtie antenna-....”

Reading this, my impression was the 1100 enhancement was compared to the bowtie antenna. But from page 6 of the Supplementary Materials, I understand that the enhancement is compared to a flat silica surface. Such ambiguity about one of the central claims of the manuscript should be avoided.

b) In Figure 1 the authors show the schematic of a three dimensional structure and in Figure 2 they discussed the field distribution over its cross section. But since the coordinate system was not clearly defined in these figures, one has to go through the text and caption trying to confirm if x and y are in the lateral and vertical directions.

c) According to the captions, Fig. 2(c) and (d) shows $|E|^2$ enhancement profiles. I believe these are plots of magnitude of $|E|^2$ from simulation, and not enhancement with respect to some other structures. If it was really enhancement (ratio of field for structure under consideration with respect to field of some other structure) the quantity on the color bar of Fig. 2(c) and y axis of Fig. 2(d) should be numbers and not $|E|^2$. Presentation of data in this way may mislead the reader.

d) In page 7 of the manuscript the authors mention

“The net enhancement experienced by a fluorophore is defined by the EM field intensity and the quantum efficiency enhancement that are discussed in Section S4, Figures S8-1 and S8-2. ”.

However, in S4 the authors use the term “quantum yield gain” instead of “quantum efficiency enhancement”. Why not use the term “quantum yield gain” in the manuscript too?

Section S4 has the following description regarding quantum yield gain

“A dipole source was placed at varied locations along the x or y axis to monitor the radiation from the dipole and the tip structure. γ_r was obtained by measuring the transmission through a closed box around a dipole source (Eq S4-1).

$$\frac{\gamma_r}{\gamma_0} = \frac{P_{structure}}{P_0} \quad \text{“}$$

There was no discussion at all about what γ_r , γ_0 , $P_{structure}$, P_0 stand for.

The entire manuscript should be reviewed carefully to improve clarity.

Comments about claims made in the manuscript

I agree with the authors that nonuniformity of field in a bowtie antenna is a fundamental limitation, and it is commendable that the authors attempted to address this issue. However, the solution they proposed comes with many other challenges. In a real life application, one would need to have an array of these devices to do analysis of a finite volume of sample. Handling such a case with bowtie antennas are relatively straight forward. For example, it is relatively easy to have collection of many bowtie antennas on a substrate which will provide many hot spots. The entire array of bowtie antennas can be illuminated with a very simple scheme. The scheme the authors proposed on the other hand, will have a much larger footprint for the same number of hotspots and will need a more complicated tapered structures for light coupling to the tip.

This does not mean the contributions the authors made are not useful. But they should recognize and clearly discuss the limitations of their design, and provide some suggestions about how future work may address some of these limitations.

Response to the reviewers:

We indicate the reviewers' comments in bold, text from the manuscript in italics, and edited text in underlined italics.

Reviewers' comments: Reviewer #1 (Remarks to the Author):

The manuscript by Kumar et al entitled “Overcoming Evanescent Field Decay Using 3D Tapered Nanocavities for On-Chip Targeted Molecular Analysis” discusses how using an “open,” tapered, metal-insulator-metal waveguide enables the formation of a rather uniform, strongly enhanced electric field. Due to the “openness” of the cavity the enhanced electric field is accessible to e.g. analyte molecules and can therefore be used as a sensing platform.

According to the Authors, the key breakthrough reported in the submitted manuscript is the creation of an enhanced electric field which is very uniform. Hence, biosensing with this device is not limited by the inhomogeneous field and the device works, according to the Authors, just as well for a short fluorescent molecule (tag) or a long one. The Authors first present a numerical analysis of the field enhancement in the device and then use fluorescent labels of different length/height to demonstrate uniform fluorescence enhancement within the whole MIM waveguide. In general, I find little that is wrong with this manuscript/work. It is pretty well written and easy to follow/understand, however, certain statements are a little confusing and/or lack precision. In my opinion the only drawback of this manuscript is that the concepts which are its basis are already very well studied.

Response: We thank the reviewer for their constructive comments and encouraging remarks about our manuscript. We have added discussion and supporting evidence to further elucidate the major challenge in the field that is solved by our device, along with the design novelties relevant to solving this challenge. We have also added further experimental data and revised some of the calculations correcting some errors. We hope that our answers and revised discussion satisfy the reviewer's concerns.

While plasmon enhanced fluorescence has been discussed as a potential revolutionary technique for sensitive detection of single or few biomolecules, restrictions associated with size and placement of molecules have prevented this method to be widely used in the field. Various reports have outlined the impact of changing fluorophore distance with respect to efficiency of plasmonic enhancement on a nanoplasmonic substrate (*References 8 -14* in the main manuscript). Puchkova *et al.* also discuss the main challenge preventing the widespread application of a dimer nanoantenna (DN) plasmonic device for fluorescence enhancement as “positioning the biological assay of interest at the DN's hotspot”. [Puchkova, Anastasiya, et al. Nano letters 15.12 (2015): 8354-8359.]

Herein we have demonstrated that an open linear 3D-tapered geometry can uniquely solve this long-standing challenge associated with plasmon enhanced fluorescence, while providing one of the highest fluorescence enhancements reported by plasmonic nanostructures. {Table S1}.

We introduce the following concepts in this manuscript:

- a) Fabrication scheme for manufacture of a linear 3D taper nanocavity accessible to fluid flow and capture of molecules (Figure 1).

- b) Analyze the coupling of fluorophores to the confined field within the narrow tip at the end of taper. We demonstrate that the energy confinement generated by 3D-tapered waveguides can be efficiently used for enhancement of emission from molecules by analyzing the change in quantum yield (the radiative decay rate of fluorophores) within the tip leading to further improvement in fluorescence (Figure S8 1-3)
- c) First realization and demonstration of the highly uniform fluorescence emission enhancement within the 3D-tapered device for a range of molecules overcoming size and distance-dependent restrictions (Figure 4).

The introduction and discussion section have now been modified to clarify these aspects.

Page 3: “Herein, we demonstrate that 3D-tapered gap plasmon nanocavity can overcome a long-standing limitation for plasmon-enhanced fluorescence demonstrating powerful emission enhancement independent of the size or position of the molecules within the nanocavity. The 3D-taper results in confinement of the electromagnetic field collected throughout the body of the device into a tiny cavity ~ 3300× smaller in volume for an $|E|^2$ enhancement close to 500. Furthermore, our analysis reveals that the 3D-taper geometry improves the coupling of molecular emitters to the electromagnetic field, delivering up to 8-28% improvement in the radiative decay rates and thus leading to even stronger enhancement of fluorescence. We specifically trap and observe single antibody molecules or arrays of molecular assemblies within the device, as well as detect low concentration (10 pM) protein molecules diffusing in solution. Significantly, optimizing the taper angle and a narrow final tip geometry of the device results in generation of a highly uniform electromagnetic field volume at the hotspot (40× improvement in uniformity compared to a bowtie nanoantenna). Combination of the strong electromagnetic confinement, powerful coupling of the emitter to the confined field and a homogenous electromagnetic field volume result in obtained enhancements around 2200 (enhancement figure of merit ~ 260) for molecular heights ranging from few angstroms to 20 nanometers, which can be further extended to 50 nanometers using the current design approach. The enhancement factor (EF) was calculated with respect to non-enhancing silica substrate such as glass chips commonly used in bioassays and enhancement figure of merit quantifies device performance independent of the fluorophore quantum yield¹⁸. Overcoming the molecule placement limitation for plasmonic enhancement of fluorescence such as presented in this manuscript can finally allow this technique to be reliably and widely applicable for a broad range of biological assays including complex molecular assemblies.”

Page 9: “We demonstrate 3D-tapered gap plasmon nanocavities which provide one of the highest enhancement of fluorescence obtained by plasmonic nanostructures (EF: ~ 2200 with figure of merit ~ 260), independent of the size of the molecular assemblies used in the assay. Overcoming molecule size and placement-dependent extreme variation in plasmonic fluorescence enhancement has been a major challenge restricting widespread application of this method in bioassays. The nanostructure geometry presented in this manuscript demonstrates a way to overcome this limitation thereby improving the consistency and range of plasmon-enhanced emission for diversely sized assembly of molecules. Simultaneously, we also demonstrate capture and visualization of single antibodies at the tip as well as sensing of proteins at low concentrations (10 pM). These advantages can be readily transferred towards applications such as highly sensitive biosensing using molecular labels of varying sizes and analysis of single molecules or tightly controlled arrays of molecules for protein orientation, protein function, and biological polymer formation studies³²⁻³⁴.”

There are two key parts to making this device – the energy concentrator (taper) and an appropriately sized final MIM waveguide – are not unique. Perhaps the combination of the above mentioned elements in this particular realization is unique, but there are no new physical concepts

involved in this work. The “overcoming of the evanescent field decay” is done using a well-studied MIM waveguide. Its “[non]evanescent field” in the final, narrow part is the same practically regardless of what how the energy is coupled into it, neglecting any transient effects. It is simply given by the geometry.

Response: The challenge we solve in this manuscript is specifically a device which provides one of the highest plasmonic enhancements obtained using plasmonic nanostructures (Table S1), independent of the size of the molecules utilized in the assay (Figure 4).

While MIM waveguides have been discussed previously as energy concentrators, efficient fluorescence enhancement within a plasmonic nanocavity depends on both the energy concentration at the hotspot and the coupling between the emitter and the electromagnetic field [1-3].

[1] Lakowicz, Joseph R. "Radiative decay engineering 5: metal-enhanced fluorescence and plasmon emission." *Analytical biochemistry* 337.2 (2005): 171-194.

[2] Chikkaraddy, Rohit, et al. "Single-molecule strong coupling at room temperature in plasmonic nanocavities." *Nature* 535.7610 (2016): 127.

[3] Mack, David L., et al. "Decoupling absorption and emission processes in super-resolution localization of emitters in a plasmonic hotspot." *Nature communications* 8 (2017): 14513.

In addition to introducing the fabrication scheme for an open, linearly 3D-tapered device, which allows molecule delivery to hotspot we have also analyzed the coupling of emitters within the hotspot, and realized the specific benefits provided by the tapered geometry. We found that the radiative decay of the fluorophore was improved by a further 8-28% due to the 3D-taper.

The concepts that are newly realized in our manuscript are:

- a) Fabrication scheme for manufacture of a linear 3D taper cavity accessible to fluid flow and capture of molecules (Figure 1).
- b) Analyze the coupling of fluorophores to the confined field within the narrow tip at the end of taper. We demonstrate that the energy confinement generated by 3D-tapered waveguides can be efficiently used for enhancement of emission from molecules by analyzing the change in quantum yield (the radiative decay rate of fluorophores) within the tip leading to further improvement in fluorescence (Figure S8 1-3)
- c) First realization and demonstration of the emission enhancement uniformity within the 3D-tapered device for a range of molecules (Figure 4).

We discuss in the manuscript:

Page 2 “While several reports have discussed novel nanoscale geometries that improve the confinement of EM fields leading to strong fluorescence enhancements¹⁵⁻¹⁸, engineering a hotspot that resolves the distance challenge between the emitter and the nanostructure surface has remained elusive.

Powerful fluorescence enhancement within a nanostructure independent of variation in molecule size and position can be expected to rely on several important factors: a) strong confinement of electromagnetic field b) powerful coupling of the emitter to the field for enhancement of emission and c) a hotspot geometry which generates uniform electromagnetic field distribution. At the same time, the hotspot geometry needs to be large enough for commonly used protein-protein binding assays (i.e. larger than antibodies, ~ 15 nm). Metal-insulator-metal (MIM) structures utilizing surface-plasmon-polariton (SPP)

propagation have been known to enable efficient confinement of EM energy¹⁹⁻²². Specifically, waveguides with a 3D taper that rely on adiabatic compression of the SPP mode inside the MIM gap have been shown to provide extreme volumetric nanoscale confinement of EM fields²³⁻²⁷. While extremely promising for energy confinement and transfer, the potential of 3D-tapered designs for coupling with fluorescent emitters and bioassays had remained unrealized primarily due to closed monolithic structures which prevented molecular integration with these devices. Furthermore, as electromagnetic field intensity on a plasmonic surface is maximum at the metal-dielectric interface, close packing of multiple metal-dielectric interfaces such as in very thin metal-insulator-metal (MIM) gaps, can result in integration of these multiple field profiles within the gap creating a more homogeneous field distribution. Therefore, a 3D-tapered structure provides unutilized potential towards these goals, for enabling confinement of a large amount of incident electromagnetic energy into a tiny MIM gap. In order to take advantage of the previously known as well as unexplored abilities of 3D-tapered MIM devices, we designed and fabricated a fluidic channel-like 3D-tapered gap plasmon nanocavity, allowing ready access of the nanostructures to molecules in solution (Figure 1a)."

The reviewer rightly suggests that homogeneous field distribution within the cavity is primarily due to the geometry of the tip at the end of the waveguide. However, the linear 3D-taper is necessary for efficient volumetric confinement of electromagnetic field into this tip, which has a volume ~3300 times smaller than the initial device body. Light collected throughout the body of the device is squeezed efficiently into the hotspot as a result of the 3D taper and we obtain a combination of powerful electromagnetic field confinement with uniform field distribution (both are important for an efficient sensing device). Figure 2e shows that the 3D taper helps improve the energy confinement to about an order of magnitude higher compared to a 2D-tapered tip.

Such MIM waveguides have been presented as biosensors before by, e.g. Dell'Olio et al (Design of a New Ultracompact Resonant Plasmonic Multi-Analyte Label-Free Biosensing Platform, Sensors 17, 1810 (2017)), as well as others.

Response: The reference indicated by the reviewer uses a plasmonic Bragg grating and uses SPR-based refractive index sensing for detection of molecular binding. The device geometry, analytical method and motivation is very different compared to our manuscript.

The detection region in the reference article is composed of Bragg gratings with width 90 nm and height 100 nm. Due to the wider cavities the electromagnetic confinement would be weaker and the hotspots reside at the corners of the structures as seen in Figure 5 (as opposed to merging of hotspots due to narrow cavities and resultant homogeneous field distribution in our tips).

Crucially, the reference manuscript focuses on SPR-based local refractive index change, where the sensitivity is dependent on the net mass of molecules adhering to the surface and the problems of emitter coupling, fluorophore distance and position are not applicable. SPR-based refractive index sensing is a commonly used method using commercial tools (Biacore, GE Healthcare Life Sciences, USA) and predominantly utilized for analysis of binding kinetics coefficients between proteins.

The challenge we solve in this manuscript is a device which provides extremely efficient fluorescence enhancements, and even more significantly - the ability to provide this enhancement independent of the size of the molecules utilized in the assay. While various nanostructures and MIM waveguides have been shown as energy concentrators and biosensors in the past, none of them have demonstrated their ability to solve this fundamental challenge associated with plasmon-enhanced fluorescence (to the best of our

knowledge).

I am puzzled by the “tip-length effect” phrase. The “cavity” (and here, in contrast to the Authors, I use “cavity” to refer only to the narrow, final part) is merely a MIM waveguide with a finite length so that a Fabry-Perot resonance can be used to modulate the field intensity in the hot spot (note, that the hot spot becomes confined in the third dimension). Also, while I understand the reason behind using the 3D taper, I do not think that the tapered part is all that important in achieving “3 to 4 times greater u_{A} (field uniformity)” (as stated on page 4). The uniformity is ensured by the MIM waveguide, while the role of the 3D taper is to compress the incident field (from the large MIM waveguide).

Response: We agree with the reviewer that the major role played by the 3D-taper is to compress the field into the 20 nm wide tip, thus providing about 7-times higher electromagnetic field confinement as compared to 2D-tapered cavity and approximately 50-times higher confinement compared to 20 nm tip alone (Figure 2e). The uniformity is primarily a result of geometry of the narrow tip region, as even a tip (with no taper) by itself provides less variation in electromagnetic field within the cavity compared to a bowtie structure (Figure 2e). However, our analysis also shows that tapered structures do see a further improvement (3-4 \times) in field uniformity compared to devices with no taper (Figure 2e).

The width of the tip (final narrow cavity) is maintained at 20 nm. We modulate the length of the tip and observe that shorter tip (20 nm being the shortest) results in the highest electromagnetic field enhancement as well as fluorescence enhancement (Figures 2f, 2g, 3c, 3d, S3, 4e).

Reviewer #2 (Remarks to the Author):

Reviewer Blind Comments to Author

Manuscript Number: Title: Overcoming Evanescent Field Decay Using 3D-Tapered Nanocavities for On-Chip Targeted Molecular Analysis

Recommendation: Reject (Publish in a different journal)

This paper presents a 3D tapered nanocavity which provides the enhancement factor of 1100 and the 40-fold fluorescence enhancement uniformity compared to a conventional bowtie nanoantenna. They fabricated the 3D tapered nanocavity device by using focused ion beam (FIB) milling and gold deposition on the SiO₂/Si substrate. They also measured fluorescence signal (AF-750) from the biotin-streptavidin binding using the 3D tapered nanocavity under the near-infrared LED excitation. While the paper contains some of the necessary data, it is missing experimental results and discussion for why a 3D tapered nanocavity was made and how it is substantially better than all previous works.

Response: We thank the reviewer for their constructive comments which have helped us improve the depth of description and clarity in our manuscript. We have added several experimental data recommended by them as well as discussion pertaining to the novelty and application of the 3D-tapered cavity. We have also revised the enhancement calculations adding new analysis and correcting errors. We hope that our revisions answer their concerns satisfactorily.

- 1. The novelty of a 3D tapered nanocavity is not clear. The 3D tapered nanocavity structure is basically same as the previously reported 3D tapered metal-insulator-metal waveguide (Choo et al. Nature Photonics, 2012). Please clearly deliver the novelty of this structure (3D tapered nanocavity) compared to previous SPP or LSPR sensors in terms of device and performance (E-field enhancement, fluorescence signal enhancement, etc.).**

Response – We have now expanded on the novelty in the manuscript and added comparisons with respect to other devices presented in the past (Table S1).

The challenge we solve in this manuscript is specifically a device which provides one of the highest enhancements obtained using plasmonic nanostructures (Table S1), and even more significantly - the ability to provide this enhancement independent of the size of the molecules utilized in the assay (Figure 4).

While MIM waveguides have been discussed previously as energy concentrators, efficient fluorescence enhancement within a plasmonic nanocavity depends on both the energy concentration at the hotspot and the coupling between the emitter and the electromagnetic field [1-3].

[1] Lakowicz, Joseph R. "Radiative decay engineering 5: metal-enhanced fluorescence and plasmon emission." *Analytical biochemistry* 337.2 (2005): 171-194.

[2] Chikkaraddy, Rohit, et al. "Single-molecule strong coupling at room temperature in plasmonic nanocavities." *Nature* 535.7610 (2016): 127.

[3] Mack, David L., et al. "Decoupling absorption and emission processes in super-resolution

localization of emitters in a plasmonic hotspot." Nature communications 8 (2017): 14513.

In addition to introducing the fabrication scheme for an open, linearly 3D-tapered device, which allows molecule delivery to hotspot we have also analyzed the coupling of emitters within the hotspot, and realized the specific benefits provided by the tapered geometry.

The concepts that are newly realized in our manuscript are:

- a) Fabrication scheme for manufacture of a linear 3D taper cavity accessible to fluid flow and capture of molecules (Figure 1).
- b) Analyze the coupling of fluorophores to the confined field within the narrow tip at the end of taper. We demonstrate that the energy confinement generated by 3D-tapered waveguides can be efficiently used for enhancement of emission from molecules by analyzing the change in quantum yield (the radiative decay rate of fluorophores) within the tip leading to further improvement in fluorescence (Figure S8 1-3)
- c) First realization and demonstration of the emission enhancement uniformity within the 3D-tapered device for a range of molecules (Figure 4).

While various nanostructures and MIM waveguides have been shown as energy concentrators and biosensors in the past, none of them have demonstrated their ability to solve this fundamental challenge associated with plasmon-enhanced fluorescence (to the best of our knowledge).

We discuss the details in manuscript as follows:

Page 2 “While several reports have discussed novel nanoscale geometries that improve the confinement of EM fields leading to strong fluorescence enhancements¹⁵⁻¹⁸, engineering a hotspot that resolves the distance challenge between the emitter and the nanostructure surface has remained elusive.

Powerful fluorescence enhancement within a nanostructure independent of variation in molecule size and position can be expected to rely on several important factors: a) strong confinement of electromagnetic field b) powerful coupling of the emitter to the field for enhancement of emission and c) a hotspot geometry which generates uniform electromagnetic field distribution. At the same time, the hotspot geometry needs to be large enough for commonly used protein-protein binding assays (i.e. larger than antibodies, ~ 15 nm). Metal-insulator-metal (MIM) structures utilizing surface-plasmon-polariton (SPP) propagation have been known to enable efficient confinement of EM energy¹⁹⁻²². Specifically, waveguides with a 3D taper that rely on adiabatic compression of the SPP mode inside the MIM gap have been shown to provide extreme volumetric nanoscale confinement of EM fields²³⁻²⁷. While extremely promising for energy confinement and transfer, the potential of 3D-tapered designs for coupling with fluorescent emitters and bioassays had remained unrealized primarily due to closed monolithic structures which prevented molecular integration with these devices. Furthermore, as electromagnetic field intensity on a plasmonic surface is maximum at the metal-dielectric interface, close packing of multiple metal-dielectric interfaces such as in very thin metal-insulator-metal (MIM) gaps, can result in integration of these multiple field profiles within the gap creating a more homogeneous field distribution. Therefore, a 3D-tapered structure provides unutilized potential towards these goals, for enabling confinement of a large amount of incident electromagnetic energy into a tiny MIM gap. In order to take advantage of the previously known as well as unexplored abilities of 3D-tapered MIM devices, we designed and fabricated a fluidic channel-like 3D-tapered gap plasmon nanocavity, allowing ready access of the nanostructures to molecules in solution (Figure 1a).”

Page 9: “We demonstrate 3D-tapered gap plasmon nanocavities which provide one of the highest enhancement of fluorescence obtained by plasmonic nanostructures (EF: ~ 2200 with figure of merit ~ 260), independent of the size of the molecular assemblies used in the assay. Overcoming molecule size and placement-dependent extreme variation in plasmonic fluorescence enhancement has been a major challenge restricting widespread application of this method in bioassays. The nanostructure geometry presented in this manuscript demonstrates a way to overcome this limitation thereby improving the consistency and range of plasmon-enhanced emission for diversely sized assembly of molecules. Simultaneously, we also demonstrate capture and visualization of single antibodies at the tip as well as sensing of proteins at low concentrations (10 pM). These advantages can be readily transferred towards applications such as highly sensitive biosensing using molecular labels of varying sizes and analysis of single molecules or tightly controlled arrays of molecules for protein orientation, protein function, and biological polymer formation studies³²⁻³⁴.”

There are parameter studies using the FDTD calculation in the supporting information. However, overall experimental studies are insufficient (Fig. 2 and Fig. 3). They only show simple experimental results for the 3D tapered nanocavity, the tip only structure, and the conventional bowtie nanoantenna. It would be better to show and discuss experimental results for the 3D tapered nanocavities with different physical dimensions as shown in the numerical results.

Response – We have added the experimental results to the manuscript now with the change in taper angle and body width (Figure S2-2). We have also added a plot showing the improved performance with respect to the tip length (Figure 4e).

Figure S2-2: Device optimization. (a) Experimental results showing that the highest fluorescence intensity from the devices was obtained for taper angle (α) close to 20° , with the body width maintained at 150 nm, tip length as 500 nm and width as 20 nm. (b) Experimental results showing that devices with body width around 150 nm had higher fluorescence emission as compared to devices with wider body widths. Device taper angle was maintained at 20° for these tests. The experimental results support the parameters indicated by simulations

Figure 4(e): Increase in mean enhancement factors with decrease in tip length to 20 nm.

2. The device coated with biotin can capture the streptavidin high concentration. What is the limit of low concentration?

Response – In terms of number of molecules, we can detect presence and emission from a single molecule at the end of taper (Figure 3d).

In terms of concentration, the detection limit is simply governed by the time taken for a single target molecule to reach the hotspot such that it is captured there. In the current model, where there is no convective flow or continuous stirring, diffusion governs the limit of detection. We were able to detect signal from 10 pM solution, which was added as a 10 μ L droplet onto the chip.

This plot has been added as Figure S7-1.

Page 6: *“Limit of detection for biomolecules on the device was examined using two types of sensing experiments. Firstly, detection of low concentration protein molecules in solution (10 pM – 1 nM) was performed, indicating device suitability for on-chip diagnostics of rare disease-specific biomarkers (Figure S7-1). The detection limit in this case is governed by diffusive transport of molecules to the plasmonic hotspot and can be accelerated by improving molecule transport using convective flow, magnetic or dielectrophoretic trapping. Secondly, we examined the capture and detection of individual or small array of molecules at the hotspot, while high concentration of molecules were present in solution. User-controlled analysis of single or small array of molecules remains an important target for high-resolution analysis of protein function and behavior³²⁻³⁴. This is specially important in cases where ligands and biomolecules are physiologically present at higher (micro – to millimolar) concentrations³⁵. We control the number of molecules captured within the tips by altering the tip length. After formation of the biotin monolayer within the 3D-tapered nanocavities, we utilized an anti-biotin IgG primary antibody (tagged with DyLight 755 ~ spectral response similar to AF-750) for performing fluorescence assays³⁶. In order to compare effect of tip length on capture of molecules, we performed these antibody binding experiments using devices with*

$l_{tip} = 500 \text{ nm}$ as well as $l_{tip} = 20 \text{ nm}$ (Figure 3c, d). The dimensions of a 3D-tapered nanocavity with a short tip ($l_{tip} = 20 \text{ nm}$, $w_{tip} = 20 \text{ nm}$) compared to those of an IgG antibody (length $\sim 15 - 20 \text{ nm}$)^{36,37} indicate that a single antibody should be specifically trapped within the tip region. Our AFM studies demonstrated uniform monolayer of PEG-biotin and antibodies on flat silica surfaces and indicated that only one antibody should specifically occupy the $20 \text{ nm} \times 20 \text{ nm}$ tip (Figure S7-2).”

Figure S7-1: Detection of low concentration molecules on 3D-tapered waveguides (tip length 500 nm). (a) Log-log plot showing increase in signal with increase in concentration of added Streptavidin (10 pM – 1000 pM). (b) Difference in signal obtained from two devices after addition of 10 pM streptavidin-AF 750. Negative control device had no biotin layer.

3. **They demonstrated specific and single molecular capture using the nanocavity which has similar dimensions with the target molecule. In this paper, they capture the IgG antibody (length 15~20 nm) using the nanocavity with 20 nm x 20 nm size. If this single molecule capture depends on the size of nanocavity and the target molecule, additional experimental results for different sizes (at least two or three) can support the principle of single molecule capture.**

Response – For the tip size shown in the manuscript (i.e. width = 20 nm and length = 20 nm), we demonstrate single protein capture for IgG antibodies with provided molecular weight around 255 kDa and size $\sim 15 \text{ nm}$. These IgG antibodies are commonly used for bioassays and are the largest among the molecules we tested. The signal increases as expected for longer tips due to presence of more molecules and falls dramatically for tip length $\sim 10 \text{ nm}$ indicating the lack of antibody molecules in the tip (Figure S9). The difference in fluorescence intensity between tip length 20 nm and 500 nm also fits well with calculations based on energy density within the tips as shown in Figure S3b.

Figure S9: Fluorescence intensity (background subtracted) obtained using tips of various lengths after performing binding assay with AF-750.

For the current fabrication method we were limited by the FIB fabrication resolution and it was trickier to reliably perform single molecule tests with smaller proteins such as streptavidin (size ~ 5 nm) or even smaller molecules. In order to perform single molecule volumetric traps with even smaller molecules we plan on changing the fabrication protocol to e-beam or nanoimprint lithography for higher resolution small cavities.

Since the current demonstration specifically demonstrates antibody-based single molecule binding, we have made this clearer in the manuscript now. We thank the reviewer for helping us clarify this point further.

Abstract: “The 3D-tapered device provides fluorescence enhancement factors close to 2200 uniformly for various molecular assemblies ranging from few angstroms to 20 nanometers in size. Furthermore, our nanostructure allows detection of low concentration (100 pM) biomarkers as well as specific capture of single antibody molecules at the nanocavity tip for high resolution molecular binding analysis.”

Page 3: “We specifically trap and observe single antibody molecules or arrays of molecular assemblies within the device, as well as detect low concentration (10 pM) protein molecules diffusing in solution.”

Page 9: “Simultaneously, we also demonstrate capture and visualization of single antibodies at the tip as well as sensing of proteins at low concentrations (10 pM).”

In addition, the authors should carefully proofread the manuscript.

Response – We have proofread the manuscript to eliminate these errors.

For example,

-Please insert scale bars for all figures of 2D E-field intensity profiles.

Response – All field intensity profile figures have scale bars included now.

-In Figure 3, the alphabets (numbers) for sub-figures and descriptions don't match. In addition, it would be better to effectively display their results to place the results under the corresponding schematic illustrations for effective delivery of results.

Response – The descriptions and figures have been changed accordingly.

On page 4, Finite-Domain Time-Difference is incorrect terminology for FDTD. Finite Difference Time Domain (FDTD) is correct terminology.

Response – The typo has been corrected.

In Figure S4b, the direction of arrow is strange. It should be in the opposite direction.

Response – We have changed the direction of the arrow to indicate that the molecule is coming out.

Once again we would like to thank the reviewer for careful assessment of our work and helping us improve the depth and clarity of our manuscript. We hope our revised version satisfies their concerns.

Reviewer #3(Remarks to the Author):

The authors present an interesting concept to improve the fluorescent enhancement intensity using a plasmonic nanotapered cavity. The concept is novel, and will be of interest to the biosensing research community, and also to other branches of photonics reserach. The work is very thorough, and the results are convincing. However, the manuscript is not written very well, and a more balanced view of the importance of the work should be presented.

Response: We thank the reviewer for their supportive and constructive comments. We found their suggestions extremely helpful to revise our manuscript and have made changes accordingly.

Comments regarding writing and presentation of results

Some of the examples about the writing and organization are described below

a) In page 3 of the manuscript the authors mentioned

“This engineered hotspot within the device tip was used for highly efficient and uniform fluorescent enhancement – the enhancement factor up to 1100 times and 40x improvement in uniformity compared to a bowtie antenna-....”

Reading this, my impression was the 1100 enhancement was compared to the bowtie antenna. But from page 6 of the Supplementary Materials, I understand that the enhancement is compared to a flat silica surface. Such ambiguity about one of the central claims of the manuscript should be avoided.

Response – We have clarified the enhancement statements through the manuscript now. We used the standard reporting methodology of reporting the enhancement with respect to a non-enhancing substrate (with similar chemistry, in this case silica). We have now made it clear throughout the manuscript.

Page 3 “*The enhancement factor (EF) was calculated with respect to non-enhancing silica substrate such as glass chips commonly used in bioassays...*”

Page 7: “*We analyzed the enhancement of fluorescence experienced by an emitter within the tip, as compared to a non-enhancing substrate (silica or glass surface).*”

We have also updated the analysis and enhancement values in the manuscript.

b) In Figure 1 the authors show the schematic of a three dimensional structure and in Figure 2 they the discussed the field distribution over its cross section. But since the coordinate system was not clearly defined in these figures, one has to go through the text and caption trying to confirm if x and y are in the lateral and vertical directions.

Response – The coordinate system has now been defined in the images.

c) According to the captions, Fig. 2(c) and (d) shows $|E|^2$ enhancement profiles. I believe these are plots of magnitude of $|E|^2$ from simulation, and not enhancement with respect to some other structures. If it was really enhancement (ratio of field for structure under consideration with respect to field of some other structure) the quantity on the color bar of Fig. 2(c) and y axis of Fig. 2(d) should be numbers and not $|E|^2$. Presentation of data in this way may mislead the reader.

Response – We apologize for the incorrect text which has been rectified now. The axis has been represented as $|E|^2$.

d) In page 7 of the manuscript the authors mention

“The net enhancement experienced by a fluorophore is defined by the EM field intensity and the quantum efficiency enhancement that are discussed in Section S4, Figures S8-1 and S8-2.”.

However, in S4 the authors use the term “quantum yield gain” instead of “quantum efficiency enhancement”. Why not use the term “quantum yield gain” in the manuscript too?

Section S4 has the following description regarding quantum yield gain

“A dipole source was placed at varied locations along the x or y axis to monitor the radiation from the dipole and the tip structure. γ_r was obtained by measuring the transmission through a closed box around a dipole source (Eq S4-1).

$$\gamma_r = \frac{P_{structure}}{P_0}$$

There was no discussion at all about what γ_r , γ_0 , $P_{structure}$, P_0 stand for.

Response – We thank the reviewer for bringing this to our attention. We have made the terminology more uniform and added discussion about terms used in equations.

SI file, page 5: *“Normalized radiative decay rate γ_r/γ_0 was obtained by measuring the transmission through a closed box around a dipole source $P_{structure}$ and dividing it by the source power P_0 (Eq. S4-1).”*

$$\frac{\gamma_r}{\gamma_0} = \frac{P_{structure}}{P_0} \quad \text{Eq. S4-1}$$

Normalized non-radiative decay rate γ_{nr}/γ_0 was obtained by subtracting the normalized transmission through a closed box containing the 3D-tapered nanocavity tip structure P_{dipole}/P_0 from (Eq. S4-2).”

The entire manuscript should be reviewed carefully to improve clarity.

Response – We thank the reviewer for helping us improve the clarity of the manuscript. We have made changes throughout the text based on their suggestions and hope that their concerns have been taken care of satisfactorily.

Comments about claims made in the manuscript

I agree with the authors that nonuniformity of field in a bowtie antenna is a fundamental limitation, and it is commendable that the authors attempted to address this issue. However, the solution they proposed comes with many other challenges. In a real life application, one would need to have an array of these devices to do analysis of a finite volume of sample. Handling such a case with bowtie antennas are relatively straight forward. For example, it is relatively easy to have collection of many bowtie antennas on a substrate which will provide many hot spots. The entire array of bowtie antennas can be illuminated with a very simple scheme. The scheme the authors proposed on the other hand, will have a much larger footprint for the same number of hotspots and will need a more complicated tapered structures for light coupling to the tip.

This does not mean the contributions the authors made are not useful. But they should recognize and clearly discuss the limitations of their design, and provide some suggestions about how future work may address some of these limitations.

Response: We agree with the reviewer that the fabrication scheme and device length we use has currently a higher footprint that currently possible using bowtie antennas. This issue can potentially be solved by developing alternative large scale fabrication methods such as nanoimprinting, anisotropic etching of silicon etc. We have added this discussion to the manuscript.

Page 9: *“While the device geometry promises several benefits, weaknesses of the current fabrication process include low throughput using focused ion beam lithography and high footprint of the device compared to smaller nanostructures. Future improvements can target the replacement of current fabrication process with wafer-scale methods including nanoimprinting, e-beam lithography and template stripping for reproducible manufacturing of nanocavities in combination with anisotropic etching methods for the tapered portions. We may also see further improvement in device performance after*

replacing ion beam milling with alternative methods mentioned above which are known to yield smoother device surfaces, and have shown such improvements in the past¹⁸."

REVIEWERS' COMMENTS second round:

Reviewer #1 (Remarks to the Author):

The Authors have addressed all comments in a satisfactory manner, although I am still not fully convinced of the novelty aspect of the work, namely the basic physics part. It is, without a doubt, a very nice engineering or applied physics device and successful realization of a taper for field enhancement and a homogeneous field and thus can be published.

Reviewer #3 (Remarks to the Author):

The authors have addressed all the concerns I had about the manuscript. I recommend the manuscript should be accepted for publication in its current form.

Response to the reviewers:

We indicate the reviewers' comments in bold.

Reviewers' comments: Reviewer #1 (Remarks to the Author):

Reviewer #1 (Remarks to the Author):

The Authors have addressed all comments in a satisfactory manner, although I am still not fully convinced of the novelty aspect of the work, namely the basic physics part. It is, without a doubt, a very nice engineering or applied physics device and successful realization of a taper for field enhancement and a homogeneous field and thus can be published.

Response: We thank the reviewer for their helpful comments which helped us improve our manuscript. We are happy that we were able to satisfactorily answer their concerns and the reviewer finds our manuscript suitable for publication.

Reviewer #3(Remarks to the Author):

The authors have addressed all the concerns I had about the manuscript. I recommend the manuscript should be accepted for publication in its current form.

Response: We thank the reviewer for their help in improving the manuscript and are glad that they find the manuscript suitable for publication.